# Testing the Efficacy of RESPONSIBLEPLAY^©^: A Multi-Theory Model (MTM)-Based Intervention Protocol for Promoting Responsible Gambling Among College Students

**DOI:** 10.3390/ijerph22060858

**Published:** 2025-05-30

**Authors:** Sidath Kapukotuwa, Kavita Batra, Christopher Johansen, Manoj Sharma

**Affiliations:** 1Department of Social and Behavioral Health, School of Public Health, University of Nevada, Las Vegas, NV 89119, USA; christopher.johansen@unlv.edu (C.J.); manoj.sharma@unlv.edu (M.S.); 2Department of Medical Education, Kirk Kerkorian School of Medicine at UNLV, University of Nevada, Las Vegas, NV 89106, USA; kavita.batra@unlv.edu; 3Office of Research, Kirk Kerkorian School of Medicine at UNLV, University of Nevada, Las Vegas, NV 89106, USA; 4Department of Internal Medicine, Kirk Kerkorian School of Medicine at UNLV, University of Nevada, Las Vegas, NV 89106, USA

**Keywords:** responsible gambling, college students, multi-theory model (MTM), behavior change, problem gambling severity index (PGSI), randomized controlled trial (RCT)

## Abstract

Background: Gambling behaviors among college students are a growing public health concern, with problem gambling rates significantly higher in this population than in the general public. Aim: This study outlines the protocol for a randomized controlled trial (RCT) evaluating the efficacy of RESPONSIBLEPLAY^©^, a Multi-theory Model (MTM)-based intervention designed to promote responsible gambling behaviors. Proposed Methods: The intervention integrates six MTM constructs—participatory dialogue, behavioral confidence, changes in the physical environment, emotional transformation, practice for change, and changes in the social environment—to address the initiation and sustenance of behavior change. College students aged 21 and older, scoring 3 or higher on the Problem Gambling Severity Index (PGSI), will be randomly assigned to either the MTM-based intervention group or a traditional knowledge-based intervention group. The participants will complete surveys assessing PGSI and MTM constructs at pre-test, post-test, and eight-week follow-up. This study aims to provide evidence for the efficacy of theory-driven intervention compared to a knowledge-based approach. Conclusions: If successful, this protocol will establish a robust framework for mitigating gambling-related harm in vulnerable college populations, paving the way for scalable, evidence-based interventions in diverse settings. The findings will contribute to the development of public health strategies that integrate theoretical constructs with practical applications to address high-risk behaviors.

## 1. Introduction

Gambling behaviors among college students have become a significant public health concern, with research indicating that this demographic is particularly susceptible to gambling-related harms [1]. Globally, approximately 10.23% of college students experience problem gambling, while 6.13% meet the criteria for Gambling Disorder (GD)—a prevalence notably higher than that in the general population [2]. Several factors contribute to this heightened risk, including increased independence, peer influence, and expanded access to gambling opportunities, all of which can lead to adverse outcomes such as academic difficulties, financial hardship, and mental health challenges [3]. Despite these risks, interventions aimed at promoting responsible gambling behaviors among college students remain limited in both scope and effectiveness.

This protocol details the implementation of a randomized controlled trial (RCT) to evaluate the efficacy of RESPONSIBLEPLAY^©^, an MTM-based intervention designed to promote responsible gambling among college students. This study addresses a critical gap in the literature by comparing the MTM-based intervention with a traditional knowledge-based approach. The findings will offer valuable insights into the effectiveness of theory-driven behavioral interventions and their potential to reduce gambling-related harm within this high-risk population.

The protocol outlines the procedures for assessing the efficacy of RESPONSIBLEPLAY^©^ in fostering responsible gambling behaviors among college students. Specifically, it aims to determine whether the MTM-based approach leads to significant improvements in gambling behaviors, as measured by the Problem Gambling Severity Index (PGSI) and MTM constructs, compared with a traditional knowledge-based intervention.

By examining the factors influencing gambling behaviors in this population, this research seeks to strengthen the evidence base for designing and implementing tailored interventions to mitigate gambling-related harms. This study’s findings will inform the development of more effective, evidence-based strategies to promote responsible gambling, enhance student well-being, and potentially extend to other at-risk populations.

### Literature Review

The growing popularity of online gambling and emerging gambling platforms has further intensified gambling-related harms, posing new challenges for public health practitioners and researchers. College students frequently participate in various forms of gambling, including sports betting, lottery games, and fantasy sports, often influenced by peer networks and the accessibility of gambling opportunities in their environment [4,5,6]. While previous interventions, such as cognitive behavioral therapy (CBT) and personalized normative feedback (PNF), have shown effectiveness in reducing gambling frequency and associated harms [7,8,9], their theoretical frameworks have often been applied inconsistently, and many lack a fully integrated approach to maximize their impact. This study seeks to address these gaps by establishing a more cohesive theoretical foundation. Left unaddressed, gambling behaviors among college students can escalate into problem gambling or GD, highlighting the urgent need for theory-driven, evidence-based interventions tailored to this population.

The Multi-theory Model (MTM) of health behavior change presents a promising framework for addressing the complexities of gambling behaviors among college students. As a fourth-generation theoretical model, MTM integrates insights from multiple behavioral theories to facilitate both the initiation and maintenance of health behavior change [10,11]. By incorporating all six constructs of MTM—participatory dialogue, behavioral confidence, changes in the physical environment, emotional transformation, practice for change, and changes in the social environment—this framework offers a comprehensive approach to designing targeted interventions that promote responsible gambling behaviors. Its holistic structure effectively addresses the intricate psychological, social, and environmental factors influencing gambling behaviors in this population. By fostering both initial behavior modification and long-term adherence, MTM provides a robust foundation for developing tailored, sustainable interventions that mitigate gambling-related harm among college students. To highlight the distinct contribution of the current study, Table 1 below summarizes how the proposed intervention differs from previously published gambling interventions.

## 2. Materials and Methods

### 2.1. Study Design

This study will implement a three-week, single-blinded educational randomized controlled trial (RCT), followed by an eight-week follow-up phase (ClinicalTrials.gov number: NCT06642155). The RCT design, widely recognized as the gold standard for evaluating intervention efficacy, was selected for its ability to minimize bias and establish causal relationships between the intervention and its outcomes through randomization [12]. To ensure methodological rigor and transparency, this study will adhere to the CONSORT checklist for RCTs [13], maintaining high standards of reporting throughout the research process. Figure 1 presents the CONSORT flow diagram.

To evaluate changes in health behavior among the participants, this study will administer pre-, post-, and follow-up assessments using the Multi-theory Model (MTM) constructs [10] and the Problem Gambling Severity Index (PGSI) [14]. The participants will be randomly assigned to one of two groups: the experimental group, which will receive the RESPONSIBLEPLAY^©^ MTM-based intervention, or the comparison group, which will receive a knowledge-based intervention. Random assignment, a fundamental component of RCTs, ensures that all the participants have an equal likelihood of being placed in either group. This approach minimizes selection bias, enhances the validity of the findings [15], and helps balance both known and unknown confounders between groups. Consequently, any observed differences in outcomes can be confidently attributed to the intervention rather than to pre-existing disparities [15]. Figure 2 presents an overview of the study design for evaluating the efficacy of responsible gambling interventions.

### 2.2. Study Sample

To assess the efficacy of an MTM-based intervention aimed at promoting responsible gambling among college students, participants will be recruited from the student population of a southwestern university. This sampling approach was chosen for its accessibility and the notable prevalence of gambling behaviors within this demographic [16]. Given that the legal gambling age in the United States is typically 21 [17], the eligibility criteria for this study will include the following:Enrollment as a student at the southwestern college.Age 21 or older.Proficiency in English.A score of 3 or higher on the PGSI [14].Consent to participate in this study.

Students currently undergoing treatment for gambling issues will be excluded to prevent potential confounding effects associated with concurrent interventions.

The Problem Gambling Severity Index (PGSI), a standardized tool for assessing gambling-related risk behaviors [14], will be used to screen the participants. Individuals scoring between 3 and 7 on the PGSI are classified as moderate-risk gamblers, indicating behaviors such as exceeding gambling budgets, losing track of time, or experiencing guilt over gambling [14]. A score of 8 or higher designates an individual as a problem gambler, reflecting significant gambling-related issues [14]. Therefore, students scoring 3 or above on the PGSI who meet all other eligibility criteria will be included in this study.

### 2.3. Sample Size Calculation

A power analysis was conducted using G*Power (version 3.1) [18] to estimate the required sample size for this study. Based on a Two-Way Mixed Analysis of Variance (ANOVA) with a medium effect size of 0.30, two intervention groups, three time points (pre-, post-, and follow-up), an alpha level of 0.05, and a beta of 0.20 (80% power), the analysis assumed a correlation of 0.5 among repeated measures and applied a non-sphericity correction of 1. The calculation indicated a required sample size of 20 participants per group. To account for potential attrition and missing data, the sample size was increased by 50%, resulting in a total of 60 participants, with 30 in each group.

### 2.4. Study Procedure

Participant recruitment will target college students aged 21 and older at the university. The recruitment efforts will include email invitations that introduce this study and invite eligible students to participate. The intervention sessions will be conducted in person. The participants will be randomly assigned to either the experimental group (RESPONSIBLEPLAY^©^: MTM-based intervention) or the comparison group (knowledge-based intervention) using SAS PROC SURVEYSELECT. Participant data will first be imported from an Excel file, and each participant will be assigned a unique ID. Simple random sampling (SRS) will then be conducted using PROC SURVEYSELECT, ensuring equal probability of assignment to either group. A fixed seed (12345) will be used to ensure reproducibility. The function will generate a selection indicator (Selected = 1 for the experimental group and Selected = 0 for the comparison group), which will be used to assign the participants accordingly. The final randomized dataset, including group assignments, will then be exported for implementation.

The intervention will be conducted over three weeks, with one session scheduled per week. Each session will span three days, offering multiple time slots to accommodate the participants’ schedules. This approach ensures that all the participants receive the same intervention content while allowing flexibility in attendance. The first session (Week 1) will include an orientation, completion of the pre-intervention survey, and the start of the intervention activities. The final session (Week 3) will consist of intervention activities followed by the post-intervention survey.

To enhance attendance and engagement, the participants will receive email reminders before each session. Additional reminders will be sent to the experimental group between sessions and after the intervention to reinforce the practice of responsible gambling behaviors. At the eight-week follow-up, the participants will attend a final assessment session, which will be scheduled in advance. Email reminders will also be sent to maximize participation.

The intervention content is designed to foster both the initiation and sustenance of responsible gambling behaviors. To support initiation, participatory dialogue will be encouraged through activities such as brainstorming, group discussions, interactive quizzes, and photovoice projects. Behavioral confidence will be enhanced through role-playing scenarios and interactive demonstrations of responsible gambling techniques. The construct of changes in the physical environment will focus on increasing access to resources, including informational brochures, support group contacts, and self-help tools.

To sustain responsible gambling behaviors, three key constructs are emphasized. Emotional transformation is fostered through psychodrama sessions and emotional intelligence workshops. Practice for change is reinforced through social media engagement, journaling, and the use of a responsible gambling app. Changes in the social environment are promoted by building social support networks through peer support groups and mentoring programs.

These strategies are meticulously designed to enhance participant engagement, ensure the standardized delivery of the intervention, and facilitate comprehensive data collection throughout this study. Figure 3 illustrates the MTM framework.

### 2.5. Ethical and Data Security

This study adheres to established ethical guidelines and has received approval from the Institutional Review Board (IRB). This protocol is currently underway and has been approved by a southwestern university under IRB Protocol # UNLV-2024-507, dated 14 April 2025. Informed consent will be obtained from all the participants before their involvement, and comprehensive measures will be implemented to ensure the confidentiality of participant information and the security of collected data.

To protect participant anonymity and confidentiality, this study will implement several key measures. Each participant will be assigned a unique identifier code, replacing names on all study materials to ensure data privacy across all phases. All digital data will be securely stored in password-protected files on Google Drive, with access restricted to authorized research team members. Informed consent will outline confidentiality safeguards, and no personally identifiable information will be collected. Any potentially identifiable data will be de-identified before analysis. Researchers will sign confidentiality agreements, and physical data, such as printed surveys, will be stored in locked cabinets. These measures are designed to uphold ethical standards and maintain participant trust.

### 2.6. Strategies to Prevent Cross-Group Influence

To minimize cross-group influence, this study will implement several strategies. Intervention sessions for the experimental and comparison groups will be scheduled separately, and group-specific materials will be clearly labeled and distributed exclusively to each group. The participants will be briefed on confidentiality and instructed not to discuss session content with members of the other group. The research team will actively monitor sessions to ensure adherence to these protocols and address any issues promptly. These measures aim to preserve this study’s integrity and validity.

### 2.7. Interventions

The experimental group will participate in a three-week intervention based on the MTM of health behavior change [10]. This intervention includes role-playing, psychodrama, journaling, and exploring recreational alternatives to address gambling behaviors. Participants will document their gambling-related activities, physical and emotional challenges, and achievements throughout the program. Each session will last approximately 60 min and be held once per week for three weeks. To encourage participation, the participants will receive a USD 20 gift card after each session and again at the eight-week follow-up.

Session 1: This session explores the pros and cons of gambling, emphasizing how the benefits of responsible gambling outweigh its risks. The participants work on building behavioral confidence, developing plans for responsible gambling, and identifying environmental changes that support responsible gambling practices (Appendix A).

Session 2: In this session, the participants reflect on their emotions related to gambling and explore techniques for transforming negative emotions into positive ones. They receive guidance on journaling and record-keeping as tools for self-awareness and accountability. Additionally, the participants engage in recreational activities as alternatives to gambling and identify sources of social support that encourage responsible gambling behaviors (Appendix A).

Session 3: The final session features a guest speaker from the Nevada Council on Problem Gambling, who will provide insights and motivation. The activities focus on initiating responsible gambling behaviors over the next eight weeks and strategies for sustaining them. The participants also receive practical demonstrations of techniques to implement and maintain responsible gambling behaviors long term (Appendix A).

The comparison group will participate in three knowledge-based sessions, each featuring lectures and presentations focused on gambling behavior (Appendix A). Each session will last approximately 60 min. At the conclusion of the eight-week follow-up, the participants will receive an incentive (a USD 20 gift card) as a token of appreciation and to support their continued learning. The Table 2 describes the alignment of intervention components with MTM constructs and behavioral outcomes.

### 2.8. Process Evaluation

The process evaluation for both the MTM-based and knowledge-based intervention groups will utilize tools developed based on the RQFSM model [19]. These tools include the Reach Evaluation Tool, which assesses the intervention’s ability to engage the target population; the Quality Evaluation Tool, which evaluates the delivery and content quality; and the Fidelity Evaluation Tool, ensuring that interventions are implemented as designed. Additionally, the Satisfaction Evaluation Tool will measure participant satisfaction with the intervention experience, while the Management Evaluation Tool will monitor the overall coordination and management of the intervention process. Together, these tools provide a comprehensive evaluation to assess both the implementation and impact of the intervention.

### 2.9. Instrumentation

The survey tool for this research consists of a 54-item questionnaire (Appendix A) specifically designed and refined through a meticulous two-stage review process conducted by a panel of six experts. These specialists, with expertise in MTM theory, instrumentation, and the target population, evaluated the questionnaire’s face and content validity in two rounds. They assessed the clarity, readability, and relevance of the items, leading to wording adjustments and the removal of one item from the behavioral confidence section. Following these revisions, the panel reached a unanimous agreement on the adequacy of the content and face validity for each MTM subscale. The finalized instrument achieved a Flesch Reading Ease score of 57.4 and a Flesch–Kincaid Grade Level of 8.7, confirming its suitability for the intended audience. In prior studies utilizing this MTM-based instrument among college populations, internal consistency estimates (Cronbach’s alpha) for the subscales ranged from 0.76 to 0.89, indicating acceptable to strong reliability [11].

Participants in this study must complete the 54-item survey based on the Multi-theory Model (MTM) at three time points: before the intervention (pre-test), immediately after the intervention (post-test), and during the follow-up period at eight weeks post-intervention. Prior to participation, all individuals will be required to provide informed consent by signing a consent form. This ensures that the participants understand the purpose of this study, their role, and their rights, including confidentiality and voluntary participation.

The initial questions of the survey focus on the frequency and types of gambling activities the participants had engaged in over the past 30 days. These questions are essential for establishing baseline measurements and serve as screening criteria for this study. The participants are asked to report the number of days they gambled during this period and to specify the types of gambling activities they participated in, such as lotteries, sports betting, poker, and online gambling. This information helps to categorize the participants based on their gambling behavior and ensures that the sample includes individuals who are actively engaged in gambling.

After the baseline questions, the survey includes the Problem Gambling Severity Index (PGSI) to assess the impact of gambling on various aspects of the participants’ lives over the past 12 months. This section consists of nine questions designed to evaluate the severity of gambling-related issues by examining factors such as financial strain, emotional distress, health complications, and social consequences linked to gambling behaviors. The participants rate their experiences on a scale from 0 (“Never”) to 3 (“Almost always”), providing a detailed measure of the extent to which gambling has affected their lives. This information helps classify participants as low-risk, moderate-risk, or problem gamblers based on their responses. The Problem Gambling Severity Index (PGSI) is a widely used and validated measure of gambling-related risk and severity. As a part of the Canadian Problem Gambling Index, it is commonly used in both general and college populations. The PGSI has demonstrated strong internal consistency in previous studies, with reported Cronbach’s alpha values typically ranging from 0.84 to 0.91, supporting its reliability as a measure of gambling-related risk behaviors [20].

The third section of the survey consists of 32 questions aimed at assessing the constructs of the Multi-theory Model (MTM) for both the initiation and sustenance of behavior change. These questions explore the participants’ readiness to initiate responsible gambling behaviors and the strategies they employ to maintain these behaviors over time. By focusing on the psychological, social, and environmental factors that influence behavior change, this section provides valuable insights into participants’ motivation, confidence, and commitment to adopting and sustaining responsible gambling practices. The final section of the survey collects demographic information, such as age, gender, and educational background, ensuring a thorough understanding of the participant population and enabling more accurate data analysis.

### 2.10. Data Analysis

The primary objective of this study is to evaluate the efficacy of an MTM-based intervention [10] in promoting responsible gambling behaviors among college students. The intervention is designed to help participants recognize the advantages of responsible gambling while addressing common barriers, enhance participatory dialogue, build behavioral confidence, and facilitate changes in their physical and social environments. Additionally, the intervention aims to foster emotional transformation and implement strategies for maintaining responsible gambling behavior over time. Data analysis will be conducted using SAS 9.4 (SAS Institute Inc., Cary, NC, USA), with a significance level of 0.05 for all statistical analyses to determine the effectiveness of the intervention. This will allow for a comprehensive evaluation of how the MTM-based approach impacts gambling behaviors and the sustainability of responsible gambling habits in the target population.

### 2.11. Per-Protocol Analysis

A per-protocol (PP) analysis may be conducted to evaluate the efficacy of the RESPONSIBLEPLAY^©^ MTM-based intervention under optimal conditions [21]. This analysis includes only those participants who fully comply with the intervention protocol and complete the follow-up assessments. By focusing on this subset of participants, the PP analysis aims to provide a more precise estimate of the intervention’s efficacy. Participants will be included in the PP analysis if they meet the following criteria:Absence of Major Protocol Violations: The participants must strictly adhere to the predefined inclusion criteria and avoid any significant deviations from the study protocol [21].Completion of Prescribed Intervention: Only participants who complete their assigned intervention as designed must be considered [21].Availability of Primary Outcome Data: The participants must have complete data for the primary outcome measure to be included in the analysis [21].

The per-protocol (PP) analysis seeks to confirm the efficacy of the interventions by excluding participants who fail to adhere to the protocol or do not complete this study. This approach ensures that the observed effects can be directly attributed to the interventions, focusing only on the participants who fully comply with the study design. In contrast, the intention-to-treat (ITT) analysis includes all randomized participants, regardless of their adherence to the protocol, providing a more conservative estimate of treatment efficacy. Together, these complementary methods offer a comprehensive view of the intervention’s impact under both optimal and real-world conditions [21].

The decision to conduct only a PP analysis is based on the goal of evaluating the intervention’s efficacy under ideal conditions, where adherence to the protocol is maximized. By focusing exclusively on the participants who strictly follow the study protocol, the PP analysis eliminates the confounding effects of non-adherence or protocol deviations. This approach provides a clearer understanding of the potential benefits of the RESPONSIBLEPLAY^©^ MTM-based intervention. It is particularly suitable for assessing health behavior interventions, as the impact of such interventions can be more accurately observed in a population that fully adheres to the prescribed protocol [21].

### 2.12. Descriptive Statistical Methods

Initially, descriptive statistics will be used to summarize the demographic characteristics and dependent variables of the study participants. The following statistical methods will be applied:Means and Standard Deviations: These measures will summarize the central tendency and variability of metric demographic and dependent variables, providing insights into their average values and dispersion [22].Frequencies and Percentages: These methods will describe the distribution of categorical data, such as the participants’ class standings and employment status, outlining the composition of the sample [22].Chi-Square Test: This test will be used to examine differences in categorical variables across independent variables. It will be applied at the pre-test to compare demographic categorical variables between the experimental and comparison groups [22].Independent Sample t-test: This test will compare the means of continuous variables across independent variables. It will be employed at the pre-test to evaluate differences in MTM constructs and other continuous variables between the experimental and comparison groups [22].

### 2.13. Inferential Statistical Methods

This study utilizes the following inferential statistical methods to evaluate the intervention’s impact as indicated in Table 3:Two-Way Mixed Analysis of Variance (ANOVA): The repeated measures analysis will be used to assess mean differences over time (pre-test, post-test, follow-up) within subjects and between the experimental and comparison groups. This test will be applied to evaluate changes in PGSI scores and MTM construct scores across the three-time points, capturing both within-subject effects (e.g., changes over time) and between-group effects (e.g., differences between the experimental and comparison groups due to the intervention) [23]. This approach allows for a thorough understanding of how the participants’ gambling behaviors and MTM constructs evolve during the course of the intervention, while accounting for the repeated nature of the measurements.Mixed Analysis of Covariance (ANCOVA): If significant baseline differences are identified in key variables (e.g., PGSI scores, MTM construct scores, or demographic factors like age or gender), an ANCOVA will be incorporated into the model to control for these differences. For instance, if the participants in one group have significantly higher baseline PGSI scores, the ANCOVA will adjust for this disparity, ensuring that the observed effects on responsible gambling behavior are attributable to the intervention rather than to pre-existing variations between groups [23]. This statistical approach enhances the precision of the findings by accounting for baseline differences, allowing for a more accurate assessment of the intervention’s impact.

## 3. Discussion

The RESPONSIBLEPLAY^©^ intervention, grounded in the Multi-theory Model (MTM) of health behavior change, offers a novel and comprehensive approach to addressing gambling-related issues among college students. By integrating the constructs of MTM, such as behavioral confidence, social support, emotional transformation, and changes in physical and social environments, this protocol is designed to address not only the knowledge gaps but also the underlying psychological and behavioral factors that drive gambling habits. It is envisaged that this approach will lead to more sustainable behavior change and better long-term outcomes for participants than traditional knowledge-based interventions, which often focus solely on increasing awareness without addressing the emotional, social, and environmental aspects of behavior change. As such, this intervention represents a significant advancement over conventional methods by offering a tailored, theory-driven framework that targets both the initiation and maintenance of responsible gambling behaviors.

### 3.1. Significance and Implications

This study addresses a critical gap in the literature by applying MTM constructs to promote responsible gambling behaviors among college students. Unlike knowledge-based interventions that often focus solely on raising awareness without facilitating lasting behavioral change, the MTM-based approach incorporates actionable strategies that aim to not only initiate but also sustain changes in gambling behavior. Through activities such as photovoice, journaling, and role-playing, the participants are equipped with practical tools to recognize, plan for, and maintain responsible gambling habits. These hands-on strategies encourage active participation and empower individuals to take ownership of their behavior, fostering long-term outcomes.

The anticipated outcomes of the RESPONSIBLEPLAY^©^ intervention include improvements in Problem Gambling Severity Index (PGSI) scores and enhanced MTM construct scores, thus validating the framework’s effectiveness for addressing gambling-related issues. By integrating both theoretical and practical components, the intervention ensures it can be adapted for diverse populations, settings, and types of gambling behaviors. For instance, strategies like limiting physical access to gambling opportunities or fostering peer support networks are flexible and applicable across various cultural and socioeconomic contexts, making the RESPONSIBLEPLAY^©^ intervention a potentially scalable and sustainable solution for addressing gambling issues on a broader scale.

### 3.2. Limitations

While this study introduces a robust framework for promoting responsible gambling behaviors, several limitations must be acknowledged regarding its implementation. First, the reliance on a convenience sample from a single southwestern university may limit the generalizability of the findings. The specific gambling behaviors and environmental influences experienced by the participants may not fully represent those of college students in other regions or demographics. To enhance the external validity of this study, future research should aim to replicate this study across diverse populations and settings to better assess its broad applicability. Additionally, the participants who complete all sessions and follow-up assessments may represent a more motivated or problem-aware subgroup, which could introduce interpretive bias. To address this concern, we will compare baseline characteristics between completers and non-completers to assess the presence of any systematic differences and report these findings transparently in future publications.

This study’s three-week intervention period, although consistent with similar behavior change interventions, may not fully capture the long-term sustainability of responsible gambling behaviors. The eight-week follow-up provides some insight into short-term changes; however, extended longitudinal studies are needed to evaluate the persistence of behavior change and the intervention’s capacity to prevent relapse over time.

Participant attrition remains a significant concern in multi-session interventions, particularly within college populations. While we increased the initial sample size by 50% to accommodate the expected dropout, we acknowledge that actual attrition may exceed this estimate. To address this, we have incorporated several retention strategies, including flexible scheduling, frequent reminders (via email), and graduated incentives to encourage session attendance and follow-up completion. We will transparently report attrition rates at each study phase and perform sensitivity analyses comparing baseline characteristics between completers and non-completers. This will help assess whether retention introduces systematic bias—particularly the concern that the participants who complete all the sessions may represent a more motivated or problem-aware subgroup. These insights will be critical in informing the design and scalability of future studies.

This study relies on the PGSI as one of the key outcome measures. While the PGSI is a widely validated and commonly used instrument, its 12-month recall period may limit its sensitivity to detecting short-term behavioral changes resulting from the intervention. To address this, we will supplement our analyses with MTM construct scores, which are specifically designed to measure short-term behavioral initiation and sustenance. Additionally, we will conduct item-level analyses of the PGSI responses (e.g., related to gambling expenditure, time lost, or emotional impact) to assess whether certain dimensions demonstrate greater sensitivity to short-term change. While the PGSI was selected for its classification utility and comparability across studies, we acknowledge that it is best suited for long-term assessments. Accordingly, we are also exploring the feasibility of a 6- to 12-month follow-up study to better align with the PGSI’s recall window and to assess sustained changes in gambling behavior. This extension will depend on participant retention and funding availability but would provide a more complete evaluation of the intervention’s longitudinal impact.

While simple randomization is used for group assignment, we acknowledge that this method may result in imbalances in baseline characteristics, particularly in smaller samples. Although reproducibility was ensured through the use of a fixed seed in SAS, future studies may benefit from using stratified or block randomization to enhance group comparability.

Lastly, the self-reported nature of gambling behaviors, while necessary for understanding personal habits and attitudes, introduces the potential for recall and social-desirability biases. The participants may underreport or overreport their gambling behaviors due to memory lapses or the desire to present themselves in a more socially acceptable light. To address these biases, future studies could incorporate objective measures or third-party reports to provide a more accurate assessment of gambling behaviors and the intervention’s impact.

### 3.3. Future Directions

The findings of this study have the potential to significantly inform both research and practice in public health interventions, particularly those focused on promoting responsible gambling behaviors. Future research could explore the integration of emerging technologies, such as mobile apps and gamified tools, to enhance participant engagement, accessibility, and long-term adherence to the intervention. For example, incorporating artificial intelligence-based tracking tools could offer real-time feedback and personalized support, allowing for the continuous monitoring of progress and tailored interventions that adapt to individual needs.

Additionally, expanding the eligibility criteria of this study to include a more diverse range of populations, such as younger students or working adults, could provide valuable insights into the intervention’s broader applicability across different age groups and socioeconomic backgrounds. This expansion would allow for a more comprehensive understanding of how the intervention performs in various contexts and life stages.

Furthermore, conducting a comparative analysis of the MTM framework with other established behavioral theories could yield a deeper understanding of the mechanisms that drive behavior change in gambling and other health-risk behaviors. By comparing MTM with theories such as Social Cognitive Theory or the Transtheoretical Model, researchers could uncover unique insights into the factors that influence responsible decision-making and sustained behavior change, potentially refining existing strategies for addressing public health challenges related to gambling.

## 4. Conclusions

The RESPONSIBLEPLAY^©^ intervention underscores the potential of theory-driven, participatory approaches in addressing gambling-related harm among college students. By integrating theoretical constructs with practical applications, this study advances the academic discourse on health behavior change and offers a replicable model for public health practitioners. The findings from this study have the potential to serve as a foundation for developing evidence-based interventions aimed at promoting responsible gambling and reducing gambling-related harm across diverse populations.

Moreover, the insights gained from this research could inform policy decisions, providing a data-driven basis for crafting policies that support gambling harm reduction. These insights could also guide future intervention designs, contributing to the creation of more tailored and effective strategies for gambling behavior change.

Furthermore, this research contributes to a broader understanding of the psychological, social, and environmental factors influencing gambling behaviors. By identifying these factors, this study fosters a more comprehensive approach to prevention and harm reduction in at-risk groups, ultimately leading to improved public health strategies and outcomes. The approach could be adapted to different contexts, broadening the reach of responsible gambling initiatives and enhancing their impact on vulnerable populations.

Beyond its theoretical significance, the proposed MTM-based intervention offers practical applicability for campus health administrators, counseling centers, and public health educators aiming to address gambling-related harm in student populations. The modular and participatory structure of the intervention makes it adaptable for group workshops, orientation programs, or digital adaptation through e-health platforms. From a programmatic perspective, the model’s focus on both initiation and maintenance constructs provides a roadmap for sustained behavior change initiatives beyond gambling potentially extending to substance use, risky sexual behavior, or digital addiction. Future research could expand this framework to more diverse student populations, incorporate longitudinal designs to assess long-term outcomes, or compare MTM with other health behavior theories (e.g., Social Cognitive Theory or Transtheoretical Model) to evaluate its relative effectiveness in various contexts.

## Figures and Tables

**Figure 1 ijerph-22-00858-f001:**
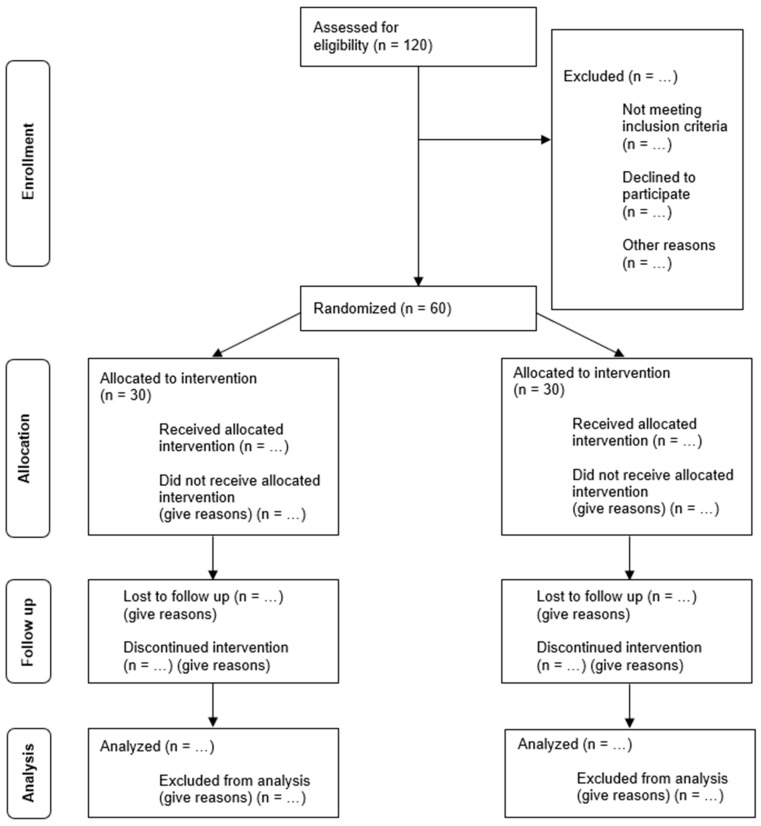
CONSORT 2010 flow diagram for the RCT [13].

**Figure 2 ijerph-22-00858-f002:**
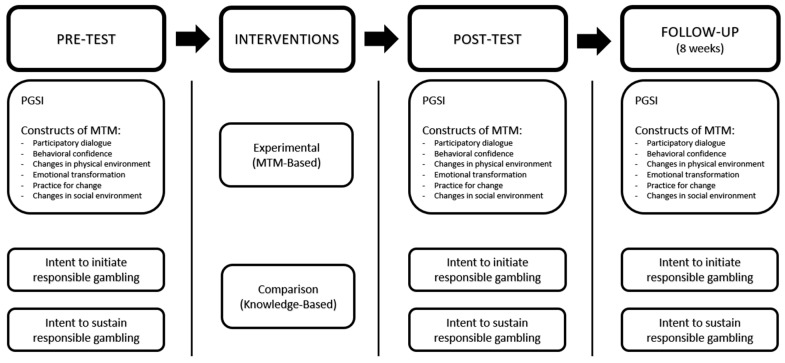
Design to test the efficacy of the responsible gambling interventions.

**Figure 3 ijerph-22-00858-f003:**
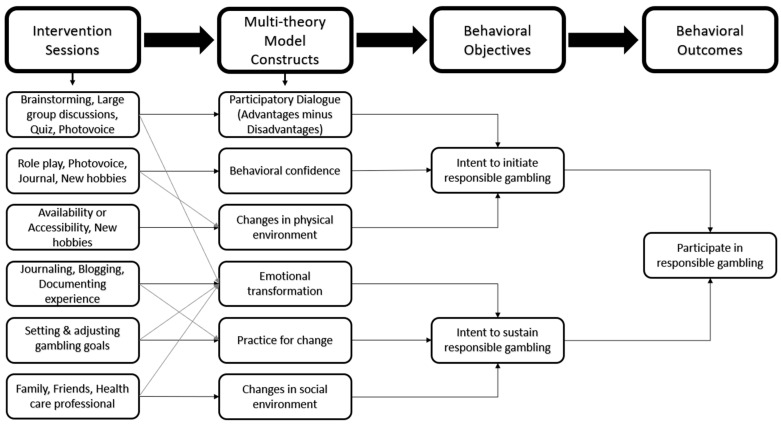
The Multi-theory Model (MTM) of the health behavior change-based intervention framework for the initiation and sustenance of responsible gambling in college students.

**Table 1 ijerph-22-00858-t001:** Comparison of current study with prior gambling interventions.

Study/Intervention	Theoretical Framework	Delivery Format	Target Group	Outcome Measures	Duration
Neighbors et al. (2015) [8]—PNF	Normative feedback	Brief online feedback	College students	Gambling frequency, risk	One time
Larimer et al. (2012) [9]—CBT + Motivational	CBT + Motivational interviewing	In-person counseling	High-risk college students	Gambling severity, coping	Multiple sessions
Pfund et al. (2023) [7]—Meta-analysis	CBT (various models)	Mixed	Mixed populations	Problem gambling severity	Varies
Current study—RESPONSIBLEPLAY^©^	Multi-theory Model (MTM)	In-person group sessions	College students (PGSI ≥ 3)	PGSI + MTM constructs	3 weeks + follow-up

PNF = personalized normative feedback; CBT = cognitive behavioral therapy.

**Table 2 ijerph-22-00858-t002:** Alignment of intervention components with MTM constructs and behavioral outcomes.

Intervention Component	MTM Construct Targeted	Mechanism of Action	Expected Behavioral Outcome
Group discussions, photovoice, brainstorming	Participatory dialogue	Enhance understanding of pros vs. cons of responsible gambling	Increased perceived value of behavior change
Role-playing, planning activities	Behavioral confidence	Build self-efficacy to manage gambling urges	Greater confidence to engage in behavior change
Brochures, self-help apps, support resources	Changes in physical environment	Improve access to tools supporting responsible gambling	Increased use of harm-reduction strategies
Psychodrama, emotional intelligence workshops	Emotional transformation	Channel emotional energy toward positive behavioral goals	Reduced emotionally driven gambling behavior
Journaling, app-based tracking, reminders	Practice for change	Reinforce behavior through reflection and repetition	Sustained responsible gambling practice
Peer mentoring, support groups	Changes in social environment	Leverage social networks for accountability and encouragement	Social reinforcement and reduced relapse

**Table 3 ijerph-22-00858-t003:** Alignment of statistical methods with study objectives.

Statistical Method	Purpose/Objective Addressed
Descriptive statistics (means, SDs)	Summarize continuous variables such as age and baseline MTM/PGSI scores.
Descriptive statistics (frequencies, %)	Describe categorical variables such as gender, class standing, and gambling types.
Chi-Square Test	Assess baseline equivalence in categorical variables between intervention and comparison groups.
Independent samples t-test	Assess baseline equivalence in continuous variables (e.g., PGSI, MTM scores) across groups.
Two-Way Mixed ANOVA	Evaluate within- and between-group differences in PGSI and MTM constructs over time (pre-, post-, follow-up), to assess efficacy of the MTM-based intervention.
Mixed ANCOVA (if needed)	Adjust for baseline differences in key variables (e.g., age, PGSI) when detected.

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
