# Peer review of "Testing the Efficacy of RESPONSIBLEPLAY^©^: A Multi-Theory Model (MTM)-Based Intervention Protocol for Promoting Responsible Gambling Among College Students"

_ijerph, 2025, doi:10.3390/ijerph22060858_

Round 1
Reviewer 1 Report
Comments and Suggestions for Authors
The present paper proposes an experimental study on the multi-theory model of promoting responsible gambling with college students being the targeted participants. There are no results as apparently the study has yet to occur.
Given that the paper was sent out for review, I am assuming that the editors are open to publishing papers without results. I find that problematic, largely because once one has the results one would assume that they would be submitted for publication. That would be double-dipping in my opinion.
That concern aside, I believe the authors have two main issues that must be addressed. The first is that I believe they have greatly underestimated the attrition rate that they are likely to see. Getting college students to show up for a single in-person laboratory session is difficult enough, but getting them to attend multiple sessions across several months will need a miracle (especially given that one of them will be a guest lecture by an "expert" - not something college students will flock to). My guess is that the majority of participants take the $20 gift card and stop after one session.
This causes three separate problems. First, it means one will need to increase the number of participants recruited. Personally, I would recommend doubling the current number. Second, this means that you will need a bigger budget, and I do not know if that is feasible. Third, and perhaps most importantly, it leaves open the possibility that small number of participants who actually make it through the entire protocol are a special population in and of themselves (perhaps those especially concerned about their gambling). That would confound interpretation of the results (beyond the noted limitation that we are already dealing with college students from a particular university).
The second main concern is the dependent measure - the PGSI. For one, it has a limited range of scores. This fact will make it difficult to detect changes. Secondly, it covers the last 12 months. Thus, even if the treatment has a positive effect, you would not expect the score on the PGSI to change much because it covers a period that extends well before the treatment. I would highly recommend that the authors consider adopting different or additional measures of gambling behavior across the treatment period. Asking about the amount of time and/or money spent on gambling in the past week would seem to be something that would introduce enough variability as to potentially identify changes.
Author Response
Given below (attached) is a point-by-point response to the reviewers' comments and concerns.

Reviewer 2 Report
Comments and Suggestions for Authors
The protocol addresses a line of research that has important consequences for people´s lives, especially those of college students. Therefore, the central theme of the protocol is considered socially relevant. Upon review, several methodological aspects that could be improved are noted:
- It is noted that simple randomization will be used; however, the literature indicates that this type of randomization has disadvantages. Therefore, it is suggested that the possibility of using another type of randomization be analyzed, such as balanced block randomization, stratified randomization, or cluster randomization.
- The protocol states that the objective is to evaluate the efficacy of an MTM-based intervention in promoting responsible gambling behaviors among college students. However, in sections 2.12. Descriptive statistical Methods, and 2.13. Inferential statistical Methods lists various statistical analyses that will be applied, but it does not specifically indicate what objective will be achieved by conducting all the analyses mentioned.
- It is proposed that the intervention to be evaluated aims to promote responsible gambling behaviors among college students; however, it is not clear how responsible gambling behavior will be achieved. This will allow the intervention to clearly observe the impact it had on the behavior under study.
Author Response

(The authors gave the same response as above.)

Reviewer 3 Report
Comments and Suggestions for Authors
Journal Name: IJERPH
Manuscript Id:
ijerph-3611776
Manuscript Title: Testing the efficacy of RESPONSIBLEPLAY©: A multi-theory model (MTM)-based intervention protocol for promoting responsible gambling among college students
This study tests the efficacy of a MTM based model for promoting responsible gambling among college students. Though the objective of this work is to establish a robust framework for gambling-related harm in vulnerable college populations, but, the model failed to find its objective. Details observations are provided below:
- The abstract section would be more concise with structured with background, methods, motivations and findings of the study.
- Add main motivation of the study into introduction to give a clear understanding of the paper. Separate the literature review from introduction. Make a table in literature review section to represent the difference between your studies with others.
- Describe why MTM is chosen in this study or why MTM is suitable for targeting responsible gambling behaviours?
- Write in details how the sample size is calculated, what is the recruitment procedure.
- Does power analysis conducted for determining the adequacy of the sample size?
- Describe clearly what is the randomization procedure?
- Describe clearly about data information regarding validity, reliability, or source.
- In my opinion, the Conclusion section should be extended. I suggest giving some managerial insights and applicability of the present model for the readers. Additionally, future extensions of the model should be more thoroughly addressed.
- Describe how do these findings compare to existing interventions in this space?
- Illustrate what are the strengths and weaknesses of the manuscript.
- Last but not the least, grammatical mistake, typos, phonetic errors should be checked from a clear point of view.
Based on my comments, I suggest for major revision of the manuscript.

Journal Name: IJERPH
Manuscript Id:
ijerph-3611776
Manuscript Title: Testing the efficacy of RESPONSIBLEPLAY©: A multi-theory model (MTM)-based intervention protocol for promoting responsible gambling among college students
This study tests the efficacy of a MTM based model for promoting responsible gambling among college students. Though the objective of this work is to establish a robust framework for gambling-related harm in vulnerable college populations, but, the model failed to find its objective. Details observations are provided below:
- The abstract section would be more concise with structured with background, methods, motivations and findings of the study.
- Add main motivation of the study into introduction to give a clear understanding of the paper. Separate the literature review from introduction. Make a table in literature review section to represent the difference between your studies with others.
- Describe why MTM is chosen in this study or why MTM is suitable for targeting responsible gambling behaviours?
- Write in details how the sample size is calculated, what is the recruitment procedure.
- Does power analysis conducted for determining the adequacy of the sample size?
- Describe clearly what is the randomization procedure?
- Describe clearly about data information regarding validity, reliability, or source.
- In my opinion, the Conclusion section should be extended. I suggest giving some managerial insights and applicability of the present model for the readers. Additionally, future extensions of the model should be more thoroughly addressed.
- Describe how do these findings compare to existing interventions in this space?
- Illustrate what are the strengths and weaknesses of the manuscript.
- Last but not the least, grammatical mistake, typos, phonetic errors should be checked from a clear point of view.
Based on my comments, I suggest for major revision of the manuscript.
Author Response

(The authors gave the same response as above.)

Round 2
Reviewer 1 Report
Comments and Suggestions for Authors
The authors have done nothing to change my concerns from the original review. I therefore cannot recommend publication.
Author Response
Given below is a point-by-point response to the reviewers' comments and concerns.
Reviewer 1 comments:
- “The present paper proposes an experimental study on the multi-theory model of promoting responsible gambling with college students being the targeted participants. There are no results as apparently the study has yet to occur. Given that the paper was sent out for review, I am assuming that the editors are open to publishing papers without results. I find that problematic, largely because once one has the results one would assume that they would be submitted for publication. That would be double-dipping in my opinion.”
Authors’ response (Round 2):
We thank the reviewer for their continued engagement and appreciate the opportunity to clarify the purpose and scope of our submission.
We respectfully note that the concern raised pertains more to the general practice of publishing study protocols than to specific issues within our manuscript. As previously indicated, this submission is an intervention protocol, submitted under the “Protocol” article type, in alignment with the International Journal of Environmental Research and Public Health’s editorial guidelines, which explicitly welcome such manuscripts.
Publishing protocols prior to data collection plays a critical role in advancing open science by promoting transparency, reproducibility, and methodological rigor. For randomized controlled trials—particularly those involving behavioral interventions—protocol papers help reduce selective reporting, increase accountability, and facilitate replication, especially when intervention details are often underreported.
To reinforce this point, we reference not only Ngcobo et al. (2025; IJERPH, 22(5), 743), but also a recent gambling-specific protocol published in BMC Pilot and Feasibility Studies:
Dobbie, F., et al. (2024). Protocol for a pilot cluster randomised controlled trial of PRoGRAM-A (Preventing Gambling-Related Harm in Adolescents): A secondary school-based social network intervention. Pilot and Feasibility Studies, 10(1), 109. https://doi.org/10.1186/s40814-024-01537-w
Moreover, entire journals such as JMIR Research Protocols are dedicated to publishing protocol papers, underscoring their value in the scientific community.
We also clarify that no data have been analyzed for this study. As such, concerns about redundant publication or “double-dipping” do not apply. Any future manuscript presenting results will explicitly reference this protocol and focus exclusively on findings and interpretation, with a clearly distinct purpose and content.
Finally, we have reinforced the protocol nature of the study in the manuscript’s title, abstract, and Introduction to avoid any potential ambiguity.
We hope this explanation addresses the reviewer’s concerns and affirms the scientific and ethical rationale for publishing this study protocol.
- “That concern aside, I believe the authors have two main issues that must be addressed. The first is that I believe they have greatly underestimated the attrition rate that they are likely to see. Getting college students to show up for a single in-person laboratory session is difficult enough, but getting them to attend multiple sessions across several months will need a miracle (especially given that one of them will be a guest lecture by an expert; - not something college students will flock to). My guess is that the majority of participants take the $20 gift card and stop after one session. This causes three separate problems. First, it means one will need to increase the number of participants recruited. Personally, I would recommend doubling the current number. Second, this means that you will need a bigger budget, and I do not know if that is feasible. Third, and perhaps most importantly, it leaves open the possibility that small number of participants who actually, make it through the entire protocol are a special population in and of themselves (perhaps those especially concerned about their gambling). That would confound interpretation of the results (beyond the noted limitation that we are already dealing with college students from a particular university).”
Authors’ response (Round 2):
We thank the reviewer for revisiting this important concern regarding potential attrition and its implications for internal validity. We fully recognize the challenge of retaining participants across multiple in-person sessions, particularly within college populations, and we take this issue seriously.
Our initial power analysis indicated that 20 participants per group were sufficient to detect a medium effect size with 80% power across three time points. To preemptively address potential dropout, we increased our planned sample size by 50%, from 40 to 60 participants. This buffer reflects both statistical prudence and practical considerations. While we acknowledge the reviewer’s concern that even higher attrition could occur, we also note that similar behavioral intervention studies have demonstrated lower-than-expected dropout. For example:
- Neighbors et al. (2015) reported retention rates of approximately 90% at both 3- and 6-month follow-ups in a randomized controlled trial of a gambling intervention for college students (J. Consult. Clin. Psychol., 83, 500–511).
- Martens et al. (2015) reported a retention rate of 94% at a 3-month follow-up in a randomized controlled trial evaluating a personalized feedback-only intervention for at-risk college student gamblers ( Consult. Clin. Psychol., 83, 494–499).
These findings suggest that high retention is possible when effective engagement and incentive strategies are in place.
In our protocol, we have implemented a number of retention strategies to maximize participant adherence:
- Multiple session times per week to increase flexibility.
- Reminder emails throughout the intervention and follow-up period.
- A graduated incentive structure (participants must attend all sessions and follow-up to receive the full set of gift cards).
We will explicitly monitor and transparently report attrition rates at each study phase (pre-test, post-test, follow-up), along with any known reasons for dropout. Additionally, we plan to conduct sensitivity analyses comparing baseline characteristics of completers and non-completers to assess for potential attrition bias.
As this is a protocol paper, one of our goals is to evaluate not only the efficacy but also the feasibility of implementing this intervention—including real-world challenges like retention. If high attrition is observed, this will inform adaptations in future trials, including potential scaling, budget planning, or multi-site expansion.
We have revised Section 3.2: Limitations to reflect a more detailed discussion of attrition risk, retention strategies, and mitigation plans. We hope this addresses the reviewer’s concern and demonstrates our thoughtful, evidence-based, and transparent approach.
- “The second main concern is the dependent measure - the PGSI. For one, it has a limited range of scores. This fact will make it difficult to detect changes. Secondly, it covers the last 12 months. Thus, even if the treatment has a positive effect, you would not expect the score on the PGSI to change much because it covers a period that extends well before the treatment. I would highly recommend that the authors consider adopting different or additional measures of gambling behavior across the treatment period. Asking about the amount of time and/or money spent on gambling in the past week would seem to be something that would introduce enough variability as to potentially identify changes.”
Authors’ response (Round 2):
We appreciate the reviewer’s continued emphasis on the suitability of our dependent measures and would like to offer a clarification. While the PGSI is an important part of our assessment strategy, it is not the sole outcome measure used to evaluate behavioral change in this study.
Our intervention is grounded in the Multi-Theory Model (MTM) of health behavior change, and we employ a validated 32-item MTM instrument that captures both initiation and sustenance of responsible gambling behaviors. Constructs such as behavioral confidence, participatory dialogue, emotional transformation, and practice for change are specifically designed to assess short-term, theory-driven behavioral shifts. These constructs are not used solely as predictors or mediators—they serve as proximal behavioral outcomes in alignment with the model’s framework.
Regarding the PGSI, we fully acknowledge its limitations—most notably the 12-month recall period, which may reduce its sensitivity to short-term change. However, the PGSI was selected for its strong psychometric properties, classification utility, and comparability across gambling studies. Its inclusion allows us to situate our findings within the broader literature on gambling severity.
We are unable to modify our IRB-approved instrument at this stage, but to strengthen behavioral outcome assessment, we will:
- Use the MTM construct scores to measure immediate behavioral engagement and readiness to change.
- Conduct item-level analyses of the PGSI to identify whether specific dimensions (e.g., time, money, guilt) show responsiveness to the intervention.
- Clearly acknowledge these limitations in interpretation, especially in the context of short-term follow-up.
- Explore the feasibility of a longer-term follow-up study (6 to 12 months) that would align with the PGSI’s recall period and better capture sustained behavior change. This extension would be contingent on retention rates and resource availability.
We have revised Section 3.2: Limitations to reflect these clarifications and forward-looking considerations. We hope this addresses the reviewer’s valid concerns and reinforces our commitment to methodological rigor and transparency.

Reviewer 3 Report
Comments and Suggestions for Authors
The authors have implemented all the comments carefully, therefore, the paper is accepted in its present form.
Author Response
Thank you very much for your feedback and they helped us improve the manuscript